# Altitudinal Patterns of Native and Invasive Alien Herbs along Roadsides in the Dayao Mountain National Nature Reserve, Guangxi, China

**Bai Li** [1,2] **, Xinying Ni** [2] **and Caiyun Zhao** [2,*]

1   School of Ecology and Environment, Zhengzhou University, Zhengzhou 450001, China
2   State Key Laboratory of Environmental Criteria and Risk Assessment, Chinese Research Academy of Environmental Sciences, Beijing 100012, China
*   Correspondence: zhaocy@craes.org.cn; Tel.: +86-138-1114-9270

**Abstract:** Invasive alien plants have rapidly established and spread in nature reserves via roads and now pose a threat to biodiversity. To understand the mechanism and distribution patterns of invasive alien herbs, we compared the altitude patterns of native and invasive alien herbs based on 105 plots in the Dayao Mountain National Nature Reserve. This study also compared the distribution patterns of new (introduced to China after 1900) and old (introduced to China before 1900) invasive alien herbs. In addition, we examined the effects of climatic factors and human activities on the distribution patterns of species richness. In our study, 151 native herbs species and 18 invasive alien herbs species were observed, of which 12 were new invasive alien herbs. Old invasive alien herbs occurred more frequently and occupied a wider range of altitudes than new invasive alien herbs. The richness of native herbs tended to decrease with increasing altitude, and the altitude patterns of the richness of all invasive herbs and new invasive alien herbs were hump-shaped. Based on an analysis using the linear mixed model, the results indicated that temperature was the main factor limiting the altitude patterns of native herbs, and that temperature and human activities were essential factors in the distribution and spread of all invasive alien herbs and new invasive alien herbs. The intensity of human interference is a crucial driver of the spread of new invasive alien herbs to higher altitudes.

**Keywords:** plant invasion; mountain; introduction time; climate factors; human activity

## 1. Introduction

Invasive alien species are one of the major factors threatening global biodiversity and ecosystem functioning [1]. In recent years, with the rapid development of trade and tourism, the rate of introduction of alien species has gradually increased [2–4]. Nature reserves are often considered protected zones for the conservation of biodiversity, ecological landscapes and ecosystems [5–7]. The high diversity of native species in nature reserves can help them compete with alien species and resist their invasion [8]. Harsh climatic conditions also prevent the invasion of alien species in nature reserves located in mountainous regions [9,10]. However, a growing number of studies [11,12] have demonstrated that invasive alien species have already invaded nature reserves and pose an even greater threat to biodiversity in nature reserves located in mountainous regions. Alien plants could invade the grassland areas from roadsides in the Glacier National Park in the United States [13]. Alien species have been observed between 2400 and 3570 m a.s.l. in protected areas of the Central Andes in Argentina [14], and 46 alien species were recorded in five protected areas of south-central Chile [15]. In China, 176 invasive alien plants have been reported in 53 national nature reserves [16]. Therefore, understanding their distribution patterns and impact factors is particularly important for the effective management of invasive alien species.

Roads are also often studied as pathways for invasive alien species because roadside habitats are usually susceptible to disturbance and present a beneficial environment for

the establishment and spread of invasive alien species [17,18]. One study compared the distribution patterns of alien species at different altitudes in 65 regions worldwide, and revealed that in most areas roadside alien species decreased with increasing altitude, with a few showing a single peak pattern [19]. In addition, at altitude gradients, roads can help species cross different climatic gradients [20]. Human activities also help invasive alien species to reach higher altitude ranges [21–23]. However, most studies in nature reserves have focused on invasive alien species at low altitudes and in island regions [24,25]. Few studies have examined higher altitudes, particularly the mountainous areas of mainland regions.

In recent years, the acceleration of the infrastructure construction process has facilitated the spread of invasive alien plants [26,27], firstly because disturbance causes damage to roadside habitats and reduces the resistance of native communities to alien plants, and secondly because human activities are gradually shifting to higher altitudes because of the continuous improvement of transportation facilities [18,28]. This phenomenon has provided an opportunity for alien plants to spread to middle and high altitudes. Therefore, human disturbance appears to be more critical than climatic factors in terms of the impact on the altitudinal patterns of alien species [29]. In addition to roads, the climate is also a key factor influencing the distribution patterns of invasive alien plants [30]. Studies have shown that the richness of invasive alien species is lower in mid- and high-altitude regions than at lower altitudes. In the harsh environment at high altitudes, it is difficult for alien species to survive. However, some studies have suggested that the warming climate may facilitate the invasion of alien species into alpine regions [31–33]. Therefore, the climate has been cited as the main factor limiting the invasion of alien species at high altitudes [34]. Compared with invasive alien species, relatively few studies have examined the effects of human disturbances on the altitudinal patterns of native species [18]. In mountainous regions, the distribution patterns of native plants at different altitudes are mainly influenced by climatic conditions and are not related to disturbances [35]. Understanding the factors limiting the distribution patterns of native and invasive alien plants can help to conserve plant diversity in mountain regions.

For invasive alien plants, the time of introduction is also a non-negligible factor [36]. The residence time (time since species introduction) is related to most of the factors that may influence the successful invasion of alien species, including the evolutionary process of the species and the occupation of a more extensive range [37]. The size of the propagule pool and opportunities for dispersal increase as alien species adapt to new environments [38]. Recent studies [39,40] have shown that land use changes caused by human activities have resulted in new alien plants spreading rapidly and shifting their distribution to higher altitudes. However, most studies have only considered the time of introduction as a factor for successful invasion, and few have investigated the differences in the distribution of altitudinal patterns of invasive alien plants in relation to different introduction times.

The Guangxi Dayao Mountain National Nature Reserve, located in central Guangxi, is globally significant for its biodiversity [41]. High mountains and complex topographies in nature reserves provide a variety of microenvironments that promote rich biodiversity [42]. To date, some investigations concerning woodland and fern species have been conducted in the reserve [41,42], but few studies have involved invasive alien species. Our aims were to answer the following questions: (1) Do the altitudinal distributional patterns differ between the native and invasive alien herbs? (2) What are the main factors that determine the distribution patterns of native and invasive herbs at altitude? (3) Are there different factors affecting the distribution patterns of old and new invasive alien herbs at altitude?

## 2. Materials and Methods

### 2.1. Study Area

The study was conducted in the Dayao Mountain National Nature Reserve, which is located in the central eastern area of Guangxi, China (110°01′–110°22′ E; 23°52′–24°22′ N). The protected area encompasses an area of 256 km$^2$ and is divided into seven zones:

Changtan River, Demei Mountain, Hekou, Huaping Mountain, Longjun Mountain, Pingzhu Mountain, and Shengtang Mountain (Figure 1). The center of the nature reserve is at high altitudes, gradually decreasing in all directions, and the geosphere is mainly composed of Cambrian sand, shale, and Devonian clastic rocks [41]. Shengtang Mountain is the central peak, which reaches 1979 m a.s.l. [42]. In this study, several areas of the reserve were surveyed along major roads, and we purposefully selected sites where invasive alien species could be reflected in the reserve. We did not set up sampling sites in areas without major roads and with low human activity levels. The main climatic feature is the significant subtropical mountain climate. The annual hours of sunshine average equal 1268.6 h, the average temperature is 17 °C, the average temperature of the hottest month (July) is 23.9 °C, and the average temperature of the coldest month (January) is 8.3 °C [41]. The annual precipitation range is 1380~2700 mm, with an annual average relative humidity rate of 83%. The forest cover in the reserve is very high, at more than 95% [43]. The main natural vegetation types are deciduous broad-leaved forests and evergreen broad-leaved forests in the low mountainous areas below 1000 m a.s.l. In the higher regions, montane evergreen broad-leaved forests and middle mountain mixed coniferous forests are widely distributed [43].

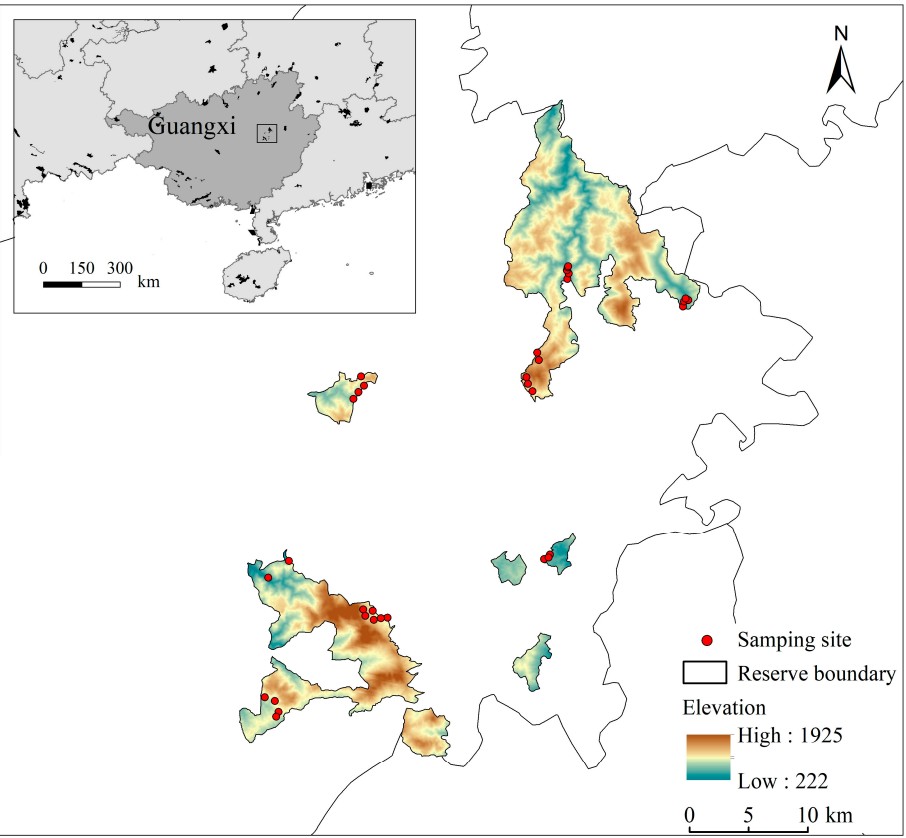

**Figure 1.** Location of the Dayao Mountain National Nature Reserve and plant survey sample sites.

## 2.2. Data Collection

The field survey was conducted in July 2021, and the sample sites were distributed across seven parts of the reserve. Thirty-three sampling sites were situated along a road that is intensively used for human activities and by vehicles, covering the altitude range between 200 and 1400 m a.s.l. The sample sites were located in the reserve's experimental area and buffer zone. The road edges were defined as consisting of distinct areas ranging from 0.5 to 4 m adjacent to the road, which are vulnerable to human activities [3]. For each sampling site, a 2 m-wide sample strip was established parallel to the road. Then, three to six 1 m × 1 m plots were randomly marked out in each sample strip, giving a total of

105 plots to be surveyed. For each plot, we recorded all of the native and invasive alien herbs and the individual numbers of each species. The invasive alien herbs and native herbs were distinguished in accordance with Ma [44]. The invasive alien herbs were identified as herbs with harmful effects in accordance with Ma [44]. The location and altitude of each plot were recorded using the Global Positioning System (GPS). Since the 19th century, the rate of introduction of invasive alien plants in China has increased rapidly, with 266 species of invasive plants having been introduced since 1900 [4]. Therefore, we classified invasive alien herbs as either old invasive alien herbs (introduced to China before 1900) or new invasive alien herbs (introduced to China after 1900) (Table S1). The introduction time of each invasive alien herb was obtained from Ma's study [44].

### 2.3. Climatic and Human Disturbance Factors

Three climatic factors were selected, namely the annual precipitation, annual mean temperature, and mean temperature of the warmest quarter, based on previous studies [45, 46]. The climate data for each site were obtained from the WorldClim data website (accessed on 23 December 2021) (https://worldclim.org) by extracting a 30-year (1970–2000) national average of meteorological data. Human disturbance was evaluated based on two indexes: (1) the distance from the plot to the nearest human settlement, which was calculated using ArcGIS; (2) the interference intensity, which was evaluated based on the habitat type around the plot (Table S2) and was estimated using a 4-grade scale: 1 = no significant interference; 2 = weak interference; 3 = moderate interference; 4 = heavy interference [47].

### 2.4. Data Analysis

We examined the potential covariance between the variables using the GGally package (Figure S1). We removed the annual mean temperature because it had strong covariance with other factors (>0.70). The native herbs, all invasive alien herbs, old invasive alien herbs, and new invasive alien herbs were the response variables. The annual precipitation, mean temperature of warmest quarter, interference intensity, and distance from the plot to the nearest human settlement were the explanatory variables. Firstly, four linear models (LM) were built and used to extensively assess the relationship between explanatory and response variables. The normality of the residuals and the homogeneity of the variance were checked using the visual inspection method for the model residuals, and the variables generally conformed to a normal distribution. Secondly, to determine the relative importance of each explanatory variable, four linear mixed models (LMM) were built based on these response variables. Climate and human disturbance factors were treated as fixed effects, and the sampling sites were treated as random effects. These variables in the mixed model were decomposed using the MuMIn and glmm.hp packages, and the contribution of each variable was calculated separately. The model analysis was performed using the lmer function of the lme4 package in R (R Core Team 2022) [48].

## 3. Results

### 3.1. Composition and Distribution of Native and Invasive Alien Herbs

In this nature reserve, a total of 151 native herb species have been recorded, belonging to 62 families and 123 genera. These species are mainly classified into Poaceae and Asteraceae, with 16 and 11 genera, respectively.

A total of 18 invasive alien herbs were observed, belonging to 8 families and 15 genera. Half of all invasive alien herbs were classified into Asteraceae. When we classified these species as old or new invasive alien herbs, the number of new invasive alien herb species was twice as large as that of old invasive alien herbs. The frequency of occurrence of old invasive alien herbs was higher than that of new invasive alien herbs. For example, the frequency rates of *Bidens pilosa*, *Erigeron canadensis*, and *Ageratum conyzoides* were 40.95%, 34.29%, and 32.38%, respectively. However, the frequency of most new invasive alien herbs was less than 10%, with the exception of *Crassocephalum crepidioides*, *Praxelis clematidea*, and *Erechtites valerianifolius* (Figure 2a). In addition, 18 invasive alien herbs were classified as

12 annual and 6 perennial invasive alien herbs. Annual invasive alien herbs occurred more frequently than perennial invasive alien herbs. The frequency of occurrence of all perennial invasive alien herbs was less than 10% (Figure 2b).

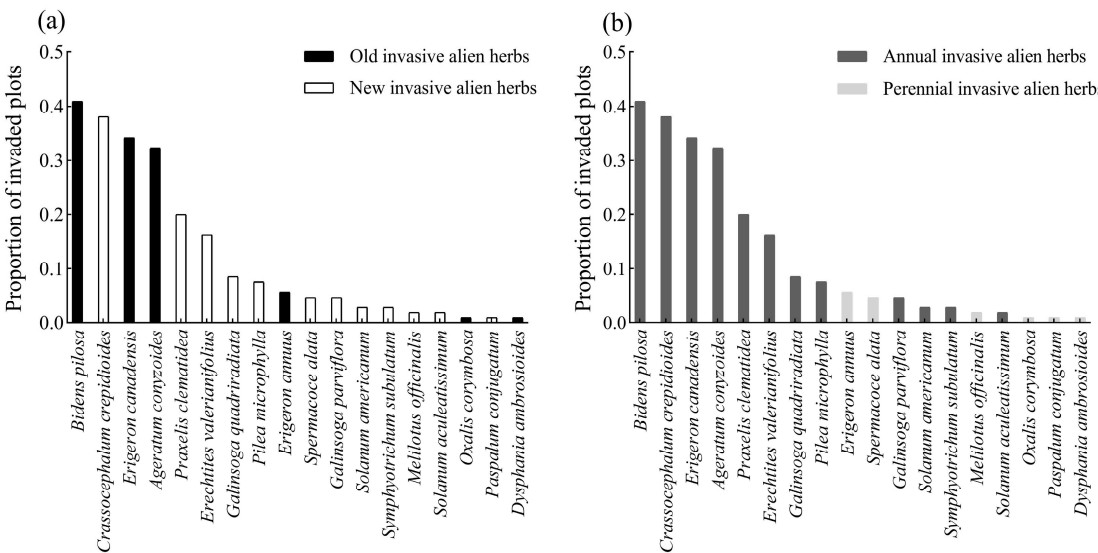

**Figure 2.** Occurrence frequency rates of (**a**) new and old invasive alien herbs and (**b**) annual and perennial invasive alien herbs in Dayao Mountain Reserve. The y-axis indicates the proportion of samples in which each species was present. Black, white, dark gray, and light gray represent old, new, annual, and perennial invasive alien herbs, respectively.

Old invasive alien herbs were found at higher altitudes and occupied a wider range of altitudes (Figure 3a). For example, *Bidens pilosa*, *Ageratum conyzoides*, and *Erigeron canadensis* were distributed in the range of 250–1300 m a.s.l. Although of the new invasive alien herbs *Crassocephalum crepidioides* was distributed in the altitude range of 300–1300 m, the other new invasive alien herbs were distributed in a narrow altitude range. Annual invasive alien herbs were distributed in the range of 250–1400 m a.s.l., whereas perennial invasive alien herbs were distributed in a smaller altitude range (Figure 3b).

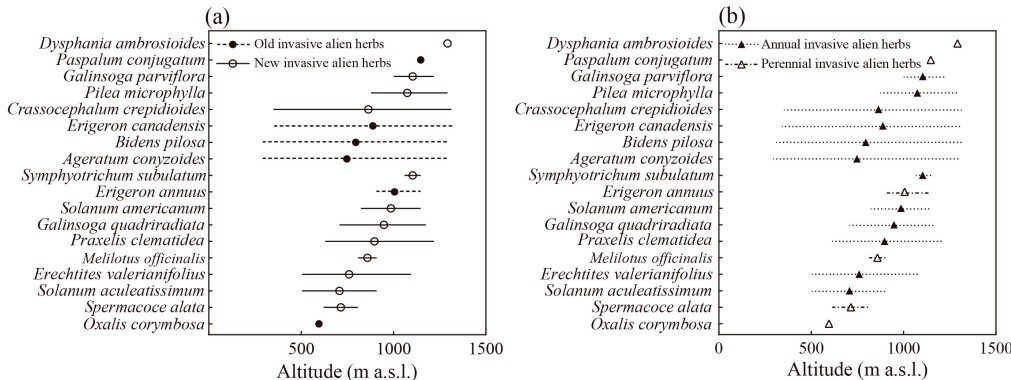

**Figure 3.** The altitudinal distribution range of (**a**) new and old invasive alien herbs and (**b**) annual and perennial invasive alien herbs. Old and new invasive alien herbs are indicated by dashed and solid lines, respectively. Annual and perennial invasive alien herbs are indicated by dotted lines and dot-dash lines, respectively. The positions of the symbols show the average altitudes at which the species occurs. Solid and hollow circles denote old and new invasive alien herbs, respectively. Solid and hollow triangles indicate annual and perennial invasive alien herbs, respectively.

### 3.2. Altitude Distribution Patterns of Native and Invasive Alien Herbs

The richness of native herbs tends to slowly decrease with increasing altitude (Figure 4a). The richness of invasive alien herbs was represented by a single hump pattern with a peak at 900 m a.s.l. (Figure 4a). The richness of new invasive alien herbs depicted a single hump pattern with a peak at 1000 m a.s.l. (Figure 4b). The richness of old invasive alien herbs was evenly distributed across altitudes (Figure 4b).

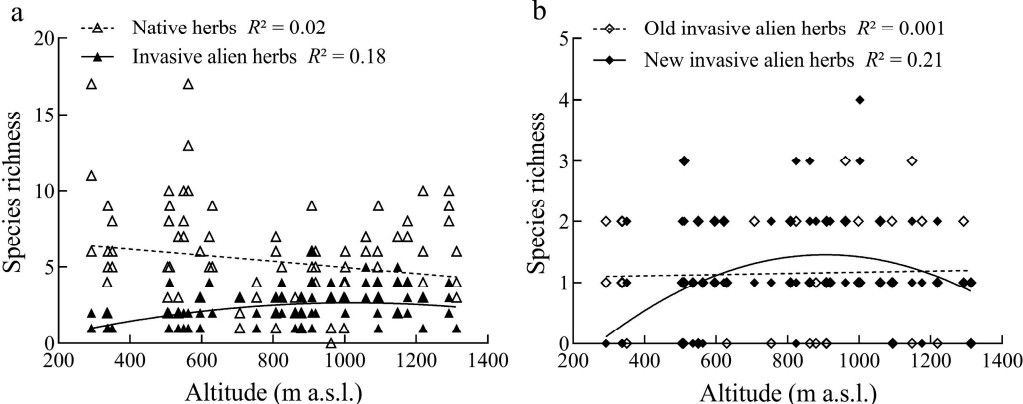

**Figure 4.** Altitudinal patterns of the richness of (**a**) native and all invasive alien herbs and (**b**) new and old invasive alien herbs in the Dayao Mountain Reserve. Hollow triangles and solid circles represent native herbs and all invasive alien herbs, respectively. Solid and open diamonds represent new and old invasive alien herbs, respectively.

### 3.3. Factors Determining the Distribution Patterns of Native and Invasive Herbs

Based on the results of the linear model analysis (Table 1), the altitude pattern of native herbs was mainly explained by the temperature (Figure 5a) and could not be correlated with human disturbance (Figure 5e).

**Table 1.** Linear models were constructed for native herbs, all invasive alien herbs, the richness levels of new and old invasive alien herbs, and various environmental factors at the altitude gradient of the reserve.

| Species Category | Environment Variables | $t$ | $p$ |
|---|---|---|---|
| Native herbs | Mean temperature of warmest quarter | 3.008 | 0.003 |
| | Annual precipitation | 0.415 | 0.679 |
| | Interference intensity | −1.038 | 0.302 |
| | Dis. human settlement | −1.060 | 0.292 |
| All invasive alien herbs | Mean temperature of warmest quarter | −2.469 | 0.015 |
| | Annual precipitation | 0.191 | 0.848 |
| | Interference intensity | 3.913 | <0.001 |
| | Dis. human settlement | −0.371 | 0.712 |
| New invasive alien herbs | Mean temperature of warmest quarter | −4.723 | <0.001 |
| | Annual precipitation | −1.733 | 0.086 |
| | Interference intensity | 3.233 | 0.002 |
| | Dis. human settlement | 0.429 | 0.668 |
| Old invasive alien herbs | Mean temperature of warmest quarter | 1.194 | 0.235 |
| | Annual precipitation | 1.767 | 0.08 |
| | Interference intensity | 1.875 | 0.05 |
| | Dis. human settlement | −0.830 | 0.408 |

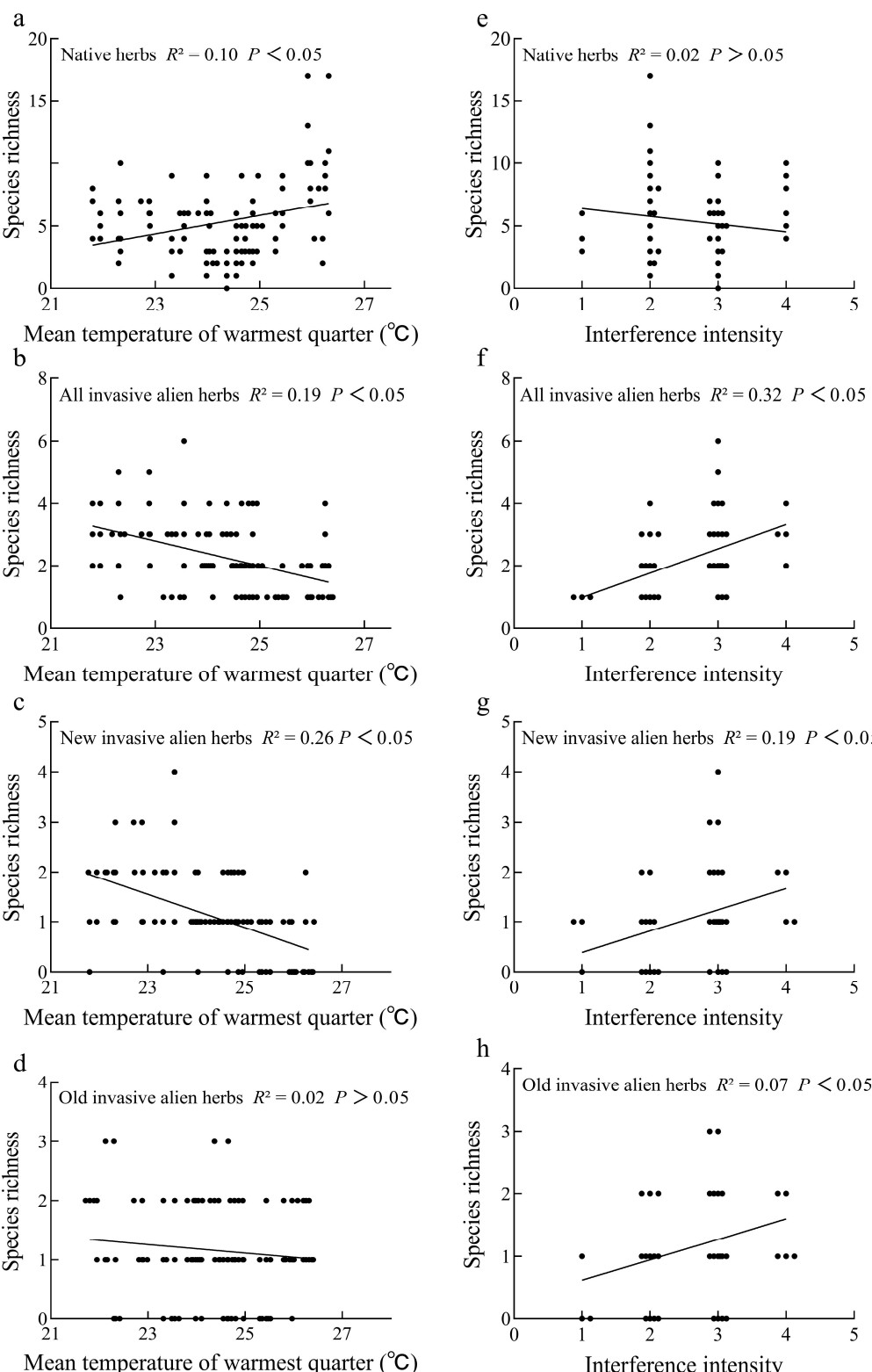

**Figure 5.** Distribution of richness for (**a**,**e**) native herbs, (**b**,**f**) all invasive alien herbs, and (**c**,**g**) new and (**d**,**h**) old invasive alien herbs with the mean temperature of the warmest quarter and interference intensity in the Dayao Mountain Reserve.

The altitude patterns of all invasive alien herbs were significantly correlated with the temperature and interference intensity (Figure 5b,f). The distribution patterns of the new invasive alien herbs were mainly explained by the temperature and interference intensity

(Figure 5c,g). The altitude patterns of old invasive alien herbs were correlated with the disturbance intensity only (Figure 5h).

Based on the results of the linear mixed model analysis (Table S3), the temperature explained 70.18% of the variation in richness of native herbs and was positively correlated with it ($p < 0.05$, Figure 6a).

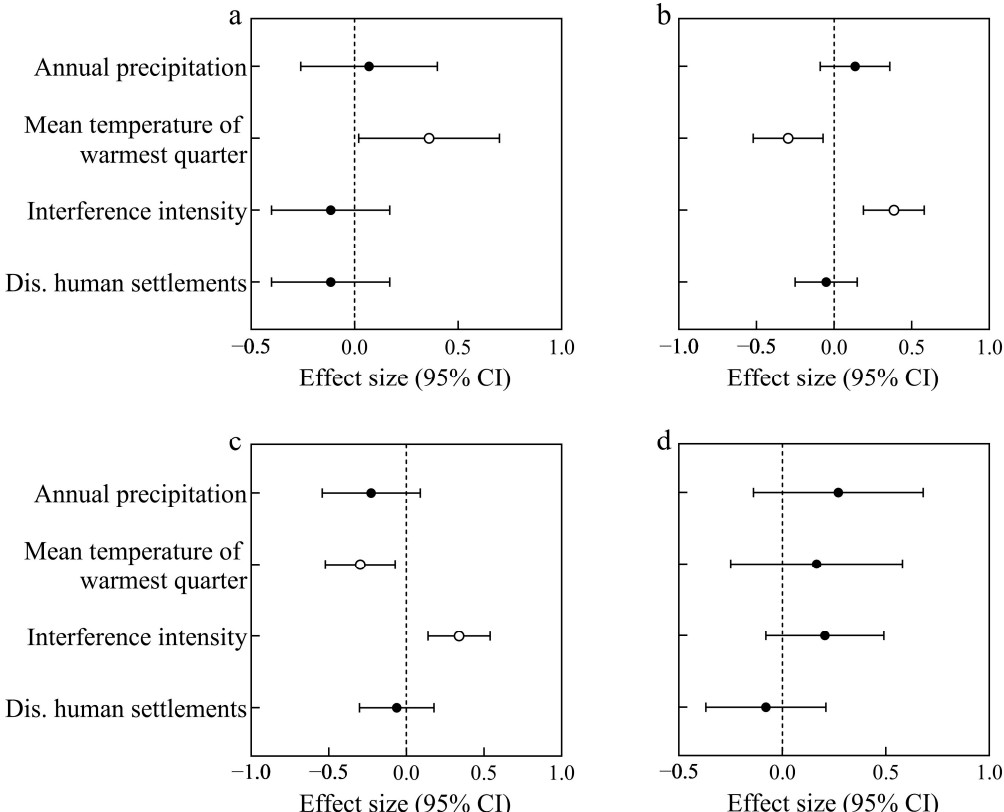

**Figure 6.** Effect values of human activities and climatic factors on (**a**) native herbs, (**b**) all invasive alien herbs, and (**c**) new and (**d**) old invasive alien herbs. Hollow circles represent species richness significantly correlated with environmental factors, solid circles represent species richness not significantly correlated with environmental factors, and dashed lines indicate an effect value of 0.

The distribution pattern of all invasive alien herbs was significantly negatively correlated with the temperature ($p < 0.05$, Figure 6b) and positively correlated with the interference intensity ($p < 0.05$, Figure 6b). The temperature explained 34.81% of the variation, but the interference intensity explained 39.71% of the variation. The richness pattern of new invasive alien herbs was mainly explained by the temperature and interference intensity (Table S3). The temperature negatively impacted the richness pattern of new invasive alien herbs and explained 53.49% of the variation ($p < 0.05$, Figure 6c), whereas the interference intensity had a positive impact and explained 34.15% of the variation ($p < 0.05$, Figure 6c). None of the environmental factors had a significant effect on the richness of old invasive alien herbs ($p > 0.05$, Figure 6d).

The annual precipitation had no significant effect on the richness of either native or invasive herbs ($p > 0.05$, Figure 6).

## 4. Discussion

### 4.1. Altitudinal Patterns of Native and Invasive Alien Herbs

This study found that invasive alien herbs in the Dayao Mountain National Nature Reserve showed a hump-shaped distribution pattern according to altitude. In contrast with our research, invasive alien herbs are currently concentrated at lower altitudes in montane

areas [47]. First, this may be in response to the arid environment at low altitudes [49] and the cold climate at high altitudes [50]. Second, our study was based on the altitudinal distribution pattern of roadside species. Invasive alien plants can occupy the remaining ecological niches because of the destruction of the original ecosystem, and then can gradually spread from lower to higher altitudes [51]. In addition, our results also suggested that the factors affecting the altitude patterns of native and invasive alien herbs were different. Native herbs have gradually adapted to the local environment through long-term development and evolution [52]. Under the influence of human activities, ecological damage may also lead to changes in the altitude patterns of native herbs [53].

In this study, differences were identified between the two invasive alien groups. Old invasive alien herbs were evenly distributed across the altitudes, whereas new invasive alien herbs showed a hump-shaped distribution pattern. The invasive alien plants with a longer residence time had a wider distribution range [54]. Becker et al. [55] showed that the altitude of invasive species increases with the time since introduction. Over time, under propagule pressure [56] and long-term genetic evolution [57], the old invasive alien plants have gradually adapted to the colder environment in order to survive at higher altitudes. In addition, human disturbance is an essential factor in the movement of species across different climatic zones [20]. Contrary to the old invasive alien herbs, the number of newly introduced aliens is increasing every year [58], even though they are not adapted to the cold environment of high altitudes [3]. Therefore, the hump-shaped distribution pattern of the new invasive alien herbs also indicated that conditions of drought at low altitudes and cold temperatures at high altitudes are unsuitable for plant survival.

*4.2. Effects of Environmental Factors on the Richness of Native and Invasive Herbs*

In recent years, the influence of environmental factors on species distribution has received extensive attention in studies of species distribution patterns across altitudinal gradients [25,59,60]. Our study revealed a significant difference in the explanatory variables between native and invasive alien herbs. The richness of native herbs was only positively correlated to temperature. This is the response of plants to climatic constraints, suggesting that higher temperatures allow for greater species richness [61]. Native plants have adapted to the local climate over tens of thousands of years of genetic evolution [62].

In contrast to native herbs, anthropogenic disturbance is an important factor affecting the richness of invasive alien plants. This is because human activities can destroy pristine ecosystems and provide more opportunities for invasive alien species [63,64]. The Dayao Mountain National Nature Reserve is a tourist destination, and tourism is a major human disturbance. With the rise of tourism, road construction has increased in reserves. Roads can expand the distribution range of alien species, acting as invasive pathways for alien species in mountainous areas [65]. The destruction of roadside habitats by pedestrians and vehicles also increases the number of suitable ecological niches for alien species [66]. In addition, we found a significant negative correlation between the richness of invasive alien herbs and temperature. This relationship suggested that mountain microenvironments provide suitable climatic conditions for the survival of invasive alien herbs and that plant invasion may be promoted in such environments [39]. Human activities play a stronger role than climatic factors in explaining the altitudinal pattern of invasive alien herbs in the Dayao Mountain National Nature Reserve.

*4.3. Effects of Environmental Factors on the Richness of Invasive Alien Herbs at Different Introduction Times*

The responses of old and new invasive alien herbs to temperature were different. The richness of old invasive alien herbs was not related to temperature, and new invasive alien herbs negatively correlated with temperature. Gasso et al. [67] showed that it took about 150 years for invasive alien plants to adapt to the local environment or to reach their maximum range. The old invasive alien herbs were all introduced to the Dayao Mountain National Nature Reserve more than 150 years ago and have gradually adapted

to the local climate conditions. Thus, they were observed at higher altitudes. Most of the new invasive alien herbs were introduced to China less than 100 years ago. For example, *Crassocephalum crepidioides* was introduced to China about 90 years ago and *Praxelis clematidea* was introduced about 40 years ago. Therefore, new invasive alien herbs were significantly correlated with temperature. Polce et al. [39] identified a negative correlation between altitudinal distribution patterns and temperature for new alien plants (>1500 years) in Europe. This is consistent with the results of our study. The mesic hypothesis stated that the microclimate is vital for new alien plants and may increase the likelihood of invasion [39,68]. This means that alien species may form a larger pool of potential species in a suitable microclimate.

Studies [69,70] have shown that the impact of human activities on the altitude pattern of invasive alien plants is more critical than climatic factors. Seebens et al. [71] showed that the global invasion of alien species has not slowed in the past few centuries because of rapid economic development and population growth. The development of commercial imports to China was accelerated by war and the ecological damage caused by the prolonged war in the 20th century, causing the introduction rate of new alien plants to peak in this period [4]. Kalwij et al. [3] showed that the movements of both vehicles and livestock are the means by which the propagules of alien species spread along roads, and have led to an annual increase in the upper altitude limit of roadside alien species, as well as in the rate of introduction of new species. Human disturbances along roadsides destroy the local vegetation composition and soil environment, reducing the biological resistance of native plant communities, creating more available resources and empty ecological niches [55], and creating opportunities for the introduction and establishment of new invasive alien species [3]. Lososova et al. [72] also confirmed that habitat changes can affect the distribution of new invasive alien plants. Thus, changes in roadside habitat can increase the establishment of new alien plants and hiking or other human activities can help them to reach higher altitude ranges [15]. Polce et al. [39] found that in Europe, old invasive alien plants were less affected by human activities than newer ones. This is consistent with the results of our study. Old invasive alien herbs generally reach the limit of their potential range after a long period of adaptation in the invaded regions [55]. New invasive alien plants, when introduced into non-native habitats, spread rapidly under the influence of human disturbance and propagule pressure, but their distribution is not uniform [40]. As a result, the number of new alien plants on the roadside has increased significantly. Unlike the new invasive alien herbs, we found that the richness of the old invasive alien herbs responded to human activities similarly to the native herbs. This phenomenon suggested that the old invasive alien herbs have been adapted over a long period and have been distributed in a wide range of altitudes after a long time of adaptation.

## 5. Conclusions

This study found that the temperature limited the altitudinal distribution patterns of native herbs, human activities facilitated the introduction and spread of alien plants, and suitable temperatures provided the conditions for alien plants to survive. Although the old invasive alien herbs occurred more frequently and occupied a more extensive altitude range than new invasive alien herbs, we found that human activities appeared to be the main driver of new invasive alien herbs in the reserve. The old invasive alien herbs gradually adapted to the local environmental conditions with increasing residence times. The new invasive alien herbs are currently distributed at low and middle altitudes because their distribution has been restricted by climatic conditions. However, new invasive alien herbs have a higher potential spreading risk under the influence of human activities. Therefore, the monitoring and prevention of newly introduced invasive alien herbs should be strengthened to reduce the proliferation risk.

**Supplementary Materials:** The following supporting information can be downloaded at: https://www.mdpi.com/article/10.3390/d15010105/s1, Figure S1: Results of correlation analysis of explanatory variables. Values above 0.7 are considered to have a strong covariance between variables; Table S1: The checklist of the alien invasive plants in Dayao Mountain National Nature Reserve, Table S2: Plots investigation of herbs in Dayao Mountain National Nature Reserve, Table S3: Linear mixed models constructed for native herbs, all invasive alien herbs, new and old invasive alien herbs richness and various environmental factors at the altitude gradient of the reserve, Table S4: Information on herbs species in plots of Dayao Mountain National Nature Reserve.

**Author Contributions:** Conceptualization, B.L. and C.Z.; methodology, B.L.; software, B.L. and X.N.; investigation, B.L. and X.N.; data curation, B.L.; writing—original draft preparation, B.L.; writing—review and editing, C.Z.; supervision, C.Z. All authors have read and agreed to the published version of the manuscript.

**Funding:** This research was supported by grants from the National Key Research and Development Program of China (2020YFC1806300) and the Central Public-Interest Scientific Institution Basal Research Fund (2022YSKY-08).

**Institutional Review Board Statement:** Not applicable.

**Data Availability Statement:** The data presented in this study are available on request from the corresponding author.

**Conflicts of Interest:** The authors declare no conflict of interest. The funders had no role in the design of the study; in the collection, analyses, or interpretation of data; in the writing of the manuscript; or in the decision to publish the results.

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
