# Peer review of "Altitudinal Patterns of Native and Invasive Alien Herbs along Roadsides in the Dayao Mountain National Nature Reserve, Guangxi, China"

_diversity, doi:10.3390/d15010105_

Round 1

Reviewer 1 Report

Dear Authors, the presented manuscript is well written study. Your results are clearly presented and then discussed. But I recommend to double-check the whole manuscript regarding grammar mistakes in the text. Somewhere, superfluous dots or commas are presented. Other parts are relatively well written. 
If I am not wrong, I have also found one more mistake. The reference of Flora of Invasive Aliens in China should be cited as Ma (2021), i.e. in the text it is like according to Ma [35]. But I may be wrong here. It is up to your choice.

Author Response

Dear Reviewer:

Thanks for your kind help to improve our MS. We have carefully considered all comments from you and revised our manuscript accordingly. We believe that our responses have well addressed all your concerns. We are very grateful for your comments and suggestions on the promotion of the article.

1: I recommend to double-check the whole manuscript regarding grammar mistakes in the text. Somewhere, superfluous dots or commas are presented. Other parts are relatively well written.

Response: Thanks for your valuable suggestion. We have corrected the above problems. And we have sent MS to language editing services to ensure the professional editing of the language.

2: If I am not wrong, I have also found one more mistake. The reference of Flora of Invasive Aliens in China should be cited as Ma (2021), i.e. in the text it is like according to Ma [35]. But I may be wrong here. It is up to your choice.

Response: Thanks for your valuable suggestion. We have reviewed according your suggestion.

Reviewer 2 Report

Dear authors, 

the manuscript is clearly written and well displayed. The theme is not very original but there are no similar articles for China. The fact that the study area is a national nature reserve gives much value to the manuscript. 

The manuscript needs some corrections that I reported in the attached file. I believe it is essential that the 105 relevé (plots) made be added to the supplements. As reported in the file I also suggest to expand the part on native species to enhance the article.

I ask in addition to the authors if they have been taken into account the type of reproduction of alien species, and if there are differences in distribution between annual and perennial alien species. It might be an interesting part to add to the article. 

In conclusion I congratulate the authors for the manuscript, I think with some changes can become a great article. 

Author Response

Dear Reviewer:

Thanks for your kind help to improve our MS. We have carefully considered all comments from you and revised our manuscript accordingly. We believe that our responses have well addressed all your concerns. We are very grateful for your comments and suggestions on the promotion of the article.

1: I believe it is essential that the 105 relevé (plots) made be added to the supplements.

Response: Thanks for your valuable suggestion. The data of 105 plots are provided in the supplementary materials as Table S4.

2: I ask in addition to the authors if they have been taken into account the type of reproduction of alien species, and if there are differences in distribution between annual and perennial alien species. 

Response: Thanks for your valuable suggestion. First, 18 invasive alien herbs were classified into three reproduction types: 13 species of sexual reproduction, 4 species of asexual reproduction and 1 species of both sexual and asexual reproduction. So, we didn’t consider to compare the distribution pattern of reproduction types of invasive alien herbs. Secondly, there are 13 species of annual herbs and 5 species of perennial herbs in our study. We added the frequency and the altitudinal distribution range of invasive alien herbs belong to two life-forms. Since the focus of this study was on the altitudinal patterns of invasive alien herbs at different introduction times, we added to the manuscript only an analysis of the frequency of occurrence and the altitudinal range of distribution of invasive alien herbs of different life-forms. The detailed discussions are provided in the manuscript as following:

In addition, 18 invasive alien herbs were classified into 13 annual and 5 perennial invasive alien herbs. Annual invasive alien herbs occurred more frequently than perennial invasive alien herbs. The frequency of occurrence of all perennial invasive alien herbs was less than 10% (Figure. 2b).

Figure 2. Occurrence frequency of invasive alien herbs in the reserve. The y-axis indicates the proportion of samples in which each species was present. Black, white, dark gray and light gray represent old, new, annual and perennial invasive alien herbs, respectively.

Annual invasive alien herbs were distributed in the range of 250–1400 m a.s.l., whereas perennial invasive alien herbs were distributed in a smaller altitude range (Figure 3b).

Figure 3. The altitudinal distribution range of invasive alien herbs, with individual points indicating only one occurrence. Old and new invasive alien herbs are indicated by dashed and solid lines, respectively. Annual and perennial invasive alien herbs are indicated by dotted lines and dot dash lines, respectively. The position of the symbols shows the average altitude at which the species occurs. Solid and hollow circles denote old and new invasive alien herbs, respectively. Solid and hollow triangles indicate annual and perennial invasive alien herbs, respectively.

3: I suggest changing the keywords that are already in the title to others.

Response: Thanks for your valuable suggestion. We have changed the keywords. The specific contents are modified as following:

plant invasion; mountain; introduction time; climate factors; human activity

4: Introduction, line 33. I suggest changing the term "barrier" perhaps "protection zone" is more correct.

Since the study was conducted in a nature reserve better explain the role of nature reserves and how they protect against the spread of alien species.

Response: Thanks for your valuable suggestion. We have reviewed as following:

In recent years, with the rapid development of trade and tourism, the rate of introduction of alien species has gradually increased [2-4]. Nature reserves are often considered protected zones for the conservation of biodiversity, ecological landscapes and ecosystems [5-7]. The high diversity of native species in nature reserves can compete with alien species and resist their invasion [8]. Harsh climatic conditions also prevent the invasion of alien species in nature reserves located in mountain regions [9,10].

5: This part of the introduction needs to be expanded a bit. I recommend adding a section on roadside alien species on other continents as well, to give a more international view to the problem.

Response: Thanks for your valuable suggestion. We have added some studies on alien species on the roadside of protected areas in other countries to the text. The detailed discussions are provided in the manuscript as following:

Alien plants could invade the grassland areas from roadsides in the Glacier national park in the United States [13]. Alien species have been observed between 2400 and 3570 m a.s.l. in protected areas of the Central Andes in Argentina [14], and 46 alien species were recorded in five protected areas of south-central Chile [15]. In China, 176 invasive alien plants have been reported in 53 national nature reserves [16].

6: Introduction, line 79 - 80. I suggest adding a reference.

Response: Thanks for your valuable suggestion. We have added references here. The detailed information is provided in the manuscript as following:

Guangxi Dayao Mountain National Nature Reserve, located in central Guangxi, is globally significant for its biodiversity [41].

7: Introduction, line 80 - 81. Unclear, formulate the sentence better.

Response: Thanks for your valuable suggestion. We have rewritten this sentence. The specific contents are modified as following:

High mountains and complex topography in nature reserves provide a variety of microenvironments that promote rich biodiversity [42].

8: I suggest to the authors to add also a brief description of the geology.

Response: Thanks for your valuable suggestion. We have added some geological information about Dayao Mountain National Nature Reserve in the manuscript. The details added to the manuscript are as following:

The center of the nature reserve is at high altitude, gradually decreasing in all directions, and the geosphere is mainly composed of Cambrian sand, shale and Devonian clastic rocks [41].

9: Study Area, line 92 – 94. Put in alphabetical order.

Response: Thanks for your valuable suggestion. We have rewritten this sentence. The specific contents are modified as following:

The protected area encompasses an area of 256 km² and is divided into seven pieces: Changtan River, Demei Mountain, Hekou, Huaping Mountain, Longjun Mountain, Pingzhu Mountain, and Shengtang Mountain.

10: Study Area, line 96 – 99. Add reference or site from which you got the information.

Response: Thanks for your valuable suggestion. We have added references here. The detailed information is provided in the manuscript as following:

The average annual sunshine hours are 1268.6 h, the average temperature is 17 ℃, the average temperature of the hottest month (July) is 23.9 ℃, and the average temperature of the coldest month (January) is 8.3 ℃. Annual precipitation is 1380~2700 mm, with an annual average relative humidity of 83% [41].

11: Given the high forest cover, it would be interesting to study whether some alien species that colonize the roadside also colonize the forest. A suggestion for future studies.

Response: Thanks for your valuable suggestion. Colleagues in our group have done some research on this idea. Although the main purpose of this study was to focus on the altitude patterns of roadside invasive alien herbs, we will also focus on this aspect in future studies.

12: I see from the figure that some areas of the reserve have not been studied, there are no sampling sites. Why? There are no roads? explain more in Study area.

Response: Thanks for your valuable suggestion. In this study, several areas of the reserve were surveyed along major roads, and we purposefully selected sites where invasive alien species could be reflected in the reserve. For the areas that were not studied, no sampling points were set up because these areas have no major roads and low human activities. We have explained the problem in the manuscript.

13: Explain better, perhaps with a figure. What is the distance between plots and the road?

Response: Thanks for your valuable suggestion. Plots for the survey were set at the edge of the roads in the reserve. Road edges are defined as consisting of distinct areas ranging from 0.5 to 4m adjacent to the road that are vulnerable to human activities [3]. We have added these to the manuscript.

14: Why did you take these three factors and not others?

Response: Thanks for your valuable suggestion. Annual precipitation, annual mean temperature and mean temperature of warmest quarter were considered as important predictors among climate variables [45,46]. These variables help us to understand and analyze the distribution patterns of invasive alien plants.

15: Data analysis, line 147. Add reference to R version.

Response: Thanks for your valuable suggestion. We have added references here. The detailed information is provided in the manuscript as following:

The model analysis is performed using the lmer function of the lme4 package in R (R Core Team 2022) [48].

16: Results, line 151-152. Poaceae is more correct. Write in italics.

Response: Thanks for your valuable suggestion. We have changed into Poaceae.

17: Unify the recognition letters of individual graphs with the style in Figure 4.

Response: Thanks for your valuable suggestion. We have adjusted Figure 5 based on the type of Figure 4. The revised figure is provided in the manuscript as following:

Figure 5. Distribution of (a, e) native herbs, (b, f) all invasive alien herbs, (c, g) new and (d, h) old invasive alien herbs richness with mean temperature of warmest quarter and interference intensity in Dayao Mountain Reserve.

18: Unify the recognition letters of individual graphs with the style in Figure 4. Enlarge the graphs and arrange them in rows of two, as in Figure 5.

Response: Thanks for your valuable suggestion. We have adjusted Figure 6 based on the type of Figure 4 and Figure 5. The revised figure is provided in the manuscript as following:

Figure. 6 Effect values of human activities and climatic factors on (a) native herbs, (b) all invasive alien herbs, (c) new and (d) old invasive alien herbs. Hollow circles represent species richness significantly correlated with environmental factors, solid circles represent species richness not significantly correlated with environmental factors, and dashed lines indicate an effect value of 0.

19: This part is interesting I suggest the authors expand it a bit. I suggest to the authors to widen the consideration on the native species found in the plots; There are endemic species? are they threatened? Are there rare species that are endangered?

Response: Thanks for your valuable suggestion. Dayao Mountain National Nature Reserve is mainly for the protection of woody plants and ferns, of which there are few herbaceous plants. Our study was conducted at the edge of the roads and with herbaceous species as the main study target. Therefore, there are no cherished and endangered species or native endemics involved within our plot information.

20: I suggest that authors expand this part to make it more robust.

Response: Thanks for your valuable suggestion. We have expanded this part of the content in the manuscript. The detailed discussions are provided in the manuscript as following:

The development of commercial imports to China was accelerated by war and the ecological damage caused by the prolonged war in the 20th century, causing the introduction rate of new alien plants to peak in this period [4]. Kalwij et al. [3] showed that both the movement of vehicles and livestock are means by which the propagules of alien species spread along roads, and have led to an annual increase in the upper altitude limit of roadside alien species as well as in the rate of introduction of new species. Human disturbance along roadsides destroys the local vegetation composition and soil environment, reduces the biological resistance of native plant communities, creates more available resources and empty ecological niches [55] and creates opportunities for the introduction and establishment of new invasive alien species [3]. Lososova et al. [72] also confirmed that habitat changes can affect the distribution of new invasive alien plants. Thus, changes in roadside habitat can increase the establishment of new alien plants and hiking or other human activities can help them to reach higher altitude ranges [15]. Polce et al. [39] found that in Europe, old invasive alien plants were less affected by human activities than newer ones. This is consistent with the results of our study. Old invasive alien herbs generally reach the limit of their potential range after a long period of adaptation in the invaded regions [55]. New invasive alien plants, when introduced into non-native habitats, spread rapidly under the influence of human disturbance and propagule pressure, but their distribution is not uniform [40].

Reviewer 3 Report

Please, consider the following suggestions:

 1.     The sentence in rows 151/152 is somehow confusing, so please re-state it anew to correct and make it clearer. When you have 27 species only out of 151 cannot state “All these species are mainly classified …” or you meant “27 genera” instead of “species” – then it will be O.K.

2.     The invasive alien herbs in Table S1 is ordered based on the Introduction time – OLD and NEW, but table still looks scrambled. An additional level of order must be applied, maybe alphabetically, so the species from one genus (like Erigeron canadensis and Erigeron annuus) are placed adjacent at least.

Author Response

Dear Reviewer:

Thanks for your kind help to improve our MS. We have carefully considered all comments from you and revised our manuscript accordingly. We believe that our responses have well addressed all your concerns. We are very grateful for your comments and suggestions on the promotion of the article.

1: The sentence in rows 151/152 is somehow confusing, so please re-state it anew to correct and make it clearer. When you have 27 species only out of 151 cannot state “All these species are mainly classified …” or you meant “27 genera” instead of “species” – then it will be O.K.

Response: Thanks for your valuable suggestion. We rearranged this sentence as following:

All these species are mainly classified into Poaceae and Asteraceae, with 16 and 11 genera, respectively.

2: The invasive alien herbs in Table S1 is ordered based on the Introduction time – OLD and NEW, but table still looks scrambled. An additional level of order must be applied, maybe alphabetically, so the species from one genus (like Erigeron canadensis and Erigeron annuus) are placed adjacent at least.

Response: Thanks for your valuable suggestion. We first ordered the species based on the time of introduction, and then, alphabetically. Species from the same family are ordered together for ease of viewing.

Table S1 has been modified in the supplementary materials as follows:

Table S1. The checklist of the alien invasive plants in Dayao Mountain National Nature Reserve.

Species name

Family

Place of origin

Introduction time

Ageratum conyzoides

Asteraceae

Tropical America

OLD

Bidens pilosa

Asteraceae

Tropical, subtropical America and Asia

OLD

Erigeron annuus

Asteraceae

North America

OLD

Erigeron canadensis

Asteraceae

North America

OLD

Dysphania ambrosioides

Amaranthaceae

Tropical America

OLD

Oxalis corymbosa

Oxalidaceae

South America

OLD

Crassocephalum crepidioides

Asteraceae

Africa

NEW

Erechtites valerianifolius

Asteraceae

Tropical America

NEW

Galinsoga parviflora

Asteraceae

South America

NEW

Galinsoga quadriradiata

Asteraceae

Mexico

NEW

Praxelis clematidea

Asteraceae

South America

NEW

Symphyotrichum subulatum

Asteraceae

North America

NEW

Melilotus officinalis

Fabaceae

Central Asia, West Asia to Southern Europe

NEW

Paspalum conjugatum

Poaceae

Tropical America

NEW

Pilea microphylla

Urticaceae

Tropical America

NEW

Solanum aculeatissimum

Solanaceae

Brazil

NEW

Solanum americanum

Solanaceae

Americas

NEW

Spermacoce alata

Rubiaceae

South America

NEW

The time of introduction refers to when the species was first discovered in China, and we distinguish between old invasive exotic herbs (introduced to China before 1900) and new invasive exotic herbs (introduced to China after 1900).

Reviewer 4 Report

The reviewed manuscript presents results of a field study, aimed at investigating the altitudinal patterns of alien invasive versus native plants in a natural reserve in southern China. Further, the authors distinguish between the alien species introduced before 1900 and after 1900. The authors conclude that i) the richness of native plants reveals a slightly negative linear trend according to the altitude; ii) alien species differ according to the residence time: old invasive aliens reveal no trend with altitude while newly introduced aliens (after 1900) reveal a hump-shaped pattern; and iii) that the altittudinal pattern of native plants is affected by mean temperature of the warmest quarter, while the altitudinal pattern of the invasive alien plants is also affected by "interference intensity". The authors conclude that the altitudinal pattern of native plants is mainly determined by environment (temperature), while the altitudinal pattern of alien plants (newly introduced aliens) is determined by human activity and that the human-induced disturbance facilitates the spread of invasive aliens to higher altitudes. This is quite far from surprising and this study does not introduce any groundshaking innovations to the ecological theory, but it does represent a decent stone to the overal mosaic of a global knowledge on the spread of invasive aliens in protected areas. However, there are few issues that need to be clarified before publication:

-          The authors state they focused on invasive aliens, however, give no reference to what they mean by this term. Did they focus on alien species in general or on invasive aliens, representing a subset of naturalized alien species? This needs to be specified.

-          The authors state that they sampled in the reserve's "buffer zone and experimental area". Do these areas really represent the vegetation of the reserve? Or do these places represent some subset of anthropogenic sites within the reserve, perhaps more related to the vegetation outside the reserve?

-          I got the impression that the predictors "distance to settlement" and "interference intensity" must be correlated somehow. Did the authors test for the correlation between these two predictors? I find it strange that "interference intensity" is listed as a significant predictor for the richness of both old and newly introduced aliens (table 1), while "distance to human settlement" is non-significant for both. Further, the authors really need to name the predictors consistently. In the Results / Factors determining…, they use "disturbance intensity". Is it the same as "interference intensity"? Please name your predictors consistently.

-          When looking at Fig 5, I can see that the "Mean temperature of the warmest quarter" is a significant predictor of the richness of both native and alien herbs – even though they show (interestingly!) the opposite patterns. However, I think that temperature is very likely negatively correlated with altitude. Did the authors attempt to separate the effects of these two factors, yielding the net effects of temperature and altitude? There are several ways how to do this, for example by running a model for one of these variables as a response and then using the residuals from this model as a response in the model testing the other variable. As it is now, the reader is left wondering wheter it is the temperature or altitude causing the presented pattern.

-          I do not really understand why the authors used linear models for testing the relation between the species richness and the selected predictors and then LMMs for estimating the explanatory power of predictors. I think that LMMs would do both, while also accounting for the autocorrelation within the data. I just do not understand this two step approach. Please explain or consider keeping only the LMMs.

-          The authors state that they recorded 151 native species and 18 aliens. This is not too many. Especially the number of native species is quite low, considering that the nature reserve ranks among the global diversity hotspots. Did the authors record all species or did they adopt some treshold of cover of biomass and therefore omitted  species with low cover of biomass? This needs to be clarified.

-          In the Abstract and Results, the authors state that the richness of invasive aliens showed hump-shaped pattern with the altitude, while the Fig. 4 suggests it is only the newly introduced aliens that show this pattern – the old-introduced aliens actually show no relation to the altitude. The Fig. 4a does show some weak hump-shaped pattern between the richness of all aliens and altitude, but is it really significant? Please make sure that i) the hump-shaped pattern is significant even when both old and new aliens are considered together and ii) that you present the results in the Abstract and Results consistently.

Besides these major issues, there are several minor issues that need to be corrected:

-          Introduction, line 31: "ecosystem functioning" rather than "ecosystem function"

-          Introduction, line 36: "mountainous regions"

-          Introduction, line 83 – 84: I suggest to re-write the n. 1) research question as follows: "Do the altitudinal distributional patterns differ between the native and invasive alien herbs?"

-          Results, line 159:  "…the frequency of most new invasive alien herbs…"

-          Results, caption to Fig 6, line 222: What does it mean "…species associated with environmental factors…"  Either explain it, re-write it or remove it.

-          Discussion, line 236 – 237: The sentence "…which may be a response to environmental adaptation…" does not make any sense to me. Either re-write it or remove it.

-          Discussion, lines 249 – 251: This sentence is not comprehensive. I suggest "Contrary to the old invasive aliens, the numbers of newly introduced aliens are increasing every year, even though they are not adapted to the cold environment of high altitudes."

-          Discussion, line 274 – 275: Either the "mesic hypothesis" or the interpretation of it does not make sense. Please re-write it or remove it.

-          Discussion, lines 298 – 299: "…and there is a variety…" Besides the grammar mistake, the second part of this sentence does not make much sense to me and could be perhaps removed.

Even though the manuscript is generally readable and comprehensive, it would strongly benefit from a careful revision of the language and stylistics.

Author Response

Dear Reviewer:

Thanks for your kind help to improve our MS. We have carefully considered all comments from you and revised our manuscript accordingly. We believe that our responses have well addressed all your concerns. We are very grateful for your comments and suggestions on the promotion of the article.

1: The authors state they focused on invasive aliens, however, give no reference to what they mean by this term. Did they focus on alien species in general or on invasive aliens, representing a subset of naturalized alien species? This needs to be specified.

Response: Thanks for your suggestion. In our study, we only focus on invasive alien herbs. We have added this in the data collection section.

2: The authors state that they sampled in the reserve's "buffer zone and experimental area". Do these areas really represent the vegetation of the reserve? Or do these places represent some subset of anthropogenic sites within the reserve, perhaps more related to the vegetation outside the reserve?

Response: Thanks for your suggestion. Since tourism and agricultural activities in the reserve are mainly concentrated in the buffer zone and experimental area, the distribution of invasive alien herbs on the roadside in the region can represent the invasion of alien herbs in the reserve to a certain extent. In addition, the core zone of the reserve is hardly to arrive.

3: I got the impression that the predictors "distance to settlement" and "interference intensity" must be correlated somehow. Did the authors test for the correlation between these two predictors? I find it strange that "interference intensity" is listed as a significant predictor for the richness of both old and newly introduced aliens (table 1), while "distance to human settlement" is non-significant for both. Further, the authors really need to name the predictors consistently. In the Results / Factors determining…, they use "disturbance intensity". Is it the same as "interference intensity"? Please name your predictors consistently.

Response: Thanks for your suggestion. We have changed into “interference intensity”. In this paper, correlation tests between independent variables were performed before constructing the model. The results of the correlation analysis are shown in the following figure, where the interference intensity and the distance to the nearest human settlement are weakly correlated.

Figure S1. Results of correlation analysis of explanatory variables. Values above 0.7 are considered to have a strong covariance between variables.

In the linear regression model, interference intensity was a significant predictor for assessing the altitudinal pattern of invasive alien herbs richness, and the distance to the nearest human settlement was independent of them. However, as the following scatterplot shows, we found a decreasing trend in the abundance of new and old invasive alien herbs with increasing distance, but they were not significantly correlated.

4: When looking at Fig 5, I can see that the "Mean temperature of the warmest quarter" is a significant predictor of the richness of both native and alien herbs – even though they show (interestingly!) the opposite patterns. However, I think that temperature is very likely negatively correlated with altitude. Did the authors attempt to separate the effects of these two factors, yielding the net effects of temperature and altitude? There are several ways how to do this, for example by running a model for one of these variables as a response and then using the residuals from this model as a response in the model testing the other variable. As it is now, the reader is left wondering wheter it is the temperature or altitude causing the presented pattern.

Response: Thanks for your valuable suggestion. A significant negative correlation between altitude and temperature was detected before running the model, and the altitude factor was removed considering that the covariance was too high. As you suggested we tried to separate the effects of altitude and temperature factors. First, we regressed altitude and invasive alien herbs richness and extracted the residual component of this model; second, we regressed the residual variables as response variables on the mean temperature of warmest quarter again and they were significantly negatively correlated (Table a). The above results are similar to the regression results of the mean temperature of warmest quarter and invasive alien herbs richness. In addition, we extracted the residuals from the regression model of invasive alien herbs richness and mean temperature of warmest quarter, and regressed the residuals as response variables with altitude again, and they were not significantly related (Table b). This result was different from the positive correlation between the richness of invasive alien herbs and altitude. In summary, temperature is an important factor influencing the altitudinal pattern of invasive alien herbs richness.

Table Regression model results for model residuals and environmental variables

Environment Variables

Residual variable

Estimate

t

P

(a)

Mean temperature of warmest quarter

- 0.1807

-2.252

0.026

(b)

Altitude

0.0003

0.791

0.431

5: I do not really understand why the authors used linear models for testing the relation between the species richness and the selected predictors and then LMMs for estimating the explanatory power of predictors. I think that LMMs would do both, while also accounting for the autocorrelation within the data. I just do not understand this two step approach. Please explain or consider keeping only the LMMs.

Response: Thanks for your valuable suggestion. Comparing the results of the linear regression model and the linear mixed-effects model, it is shown that interference intensity is a significant predictor variable of the richness of old invasive alien herbs in the linear regression model. However, after adding a random effects model to the linear mixed effects model, we found no correlation between interference intensity and the richness of the old invasive alien herbs. Therefore, it is necessary to apply two models to assess the influence of environmental factors on the richness of invasive alien herbs.

6: The authors state that they recorded 151 native species and 18 aliens. This is not too many. Especially the number of native species is quite low, considering that the nature reserve ranks among the global diversity hotspots. Did the authors record all species or did they adopt some treshold of cover of biomass and therefore omitted species with low cover of biomass? This needs to be clarified.

Response: Thanks for your suggestion. Our survey focused on the species along the road, so the data not fully reflect all species in the reserve. The number of species does not fully reflect the number of all species in the reserve because species surveys were not conducted for the understory and areas away from roads.

7: In the Abstract and Results, the authors state that the richness of invasive aliens showed hump-shaped pattern with the altitude, while the Fig. 4 suggests it is only the newly introduced aliens that show this pattern – the old-introduced aliens actually show no relation to the altitude. The Fig. 4a does show some weak hump-shaped pattern between the richness of all aliens and altitude, but is it really significant? Please make sure that i) the hump-shaped pattern is significant even when both old and new aliens are considered together and ii) that you present the results in the Abstract and Results consistently.

Response: Thanks for your valuable suggestion. Thanks for your suggestion. We have reviewed this. Although all invasive alien herbs richness showed a weak hump-shaped pattern with altitude in Fig. 4a, we found a significant correlation between them by regression model. The new invasive exotic herbs in Figure 4b showed a clear hump-shaped pattern with altitude, and there was also a significant correlation between them.

8: Introduction, line 31: "ecosystem functioning" rather than "ecosystem function"

Response: Thanks for your suggestion. We have changed according to your suggestion.

9: Introduction, line 36: "mountainous regions"

Response: Thanks for your valuable suggestion. We have changed according to your suggestion.

10: Introduction, line 83 – 84: I suggest to re-write the n. 1) research question as follows: "Do the altitudinal distributional patterns differ between the native and invasive alien herbs?"

Response: Thanks for your valuable suggestion. We have changed according to your suggestion.

11: Results, line 159:  "…the frequency of most new invasive alien herbs…"

Response: Thanks for your valuable suggestion. We have changed according to your suggestion.

12: Results, caption to Fig 6, line 222: What does it mean "…species associated with environmental factors…"  Either explain it, re-write it or remove it.

Response: Thanks for your valuable suggestion. We are sorry for our vague description. The specific contents are modified as follows:

Hollow circles represent species richness significantly correlated with environmental factors; solid circles represent species richness no significantly correlated with environmental factors.

13: Discussion, line 236 – 237: The sentence "…which may be a response to environmental adaptation…" does not make any sense to me. Either re-write it or remove it.

Response: Thanks for your valuable suggestion. We have removed this sentence.

14: Discussion, lines 249 – 251: This sentence is not comprehensive. I suggest "Contrary to the old invasive aliens, the numbers of newly introduced aliens are increasing every year, even though they are not adapted to the cold environment of high altitudes."

Response: Thanks for your valuable suggestion. We have changed according to your suggestion.

15: Discussion, line 274 – 275: Either the "mesic hypothesis" or the interpretation of it does not make sense. Please rewrite it or remove it.

Response: Thanks for your valuable suggestion. We have rewritten this sentence. The specific contents are modified as follows:

This relationship suggested that mountain microenvironments provide suitable climatic conditions for the survival of invasive alien herbs and that plant invasion may be promoted in such environments [39].

16: Discussion, lines 298 – 299: "…and there is a variety…" Besides the grammar mistake, the second part of this sentence does not make much sense to me and could be perhaps removed.

Response: Thanks for your valuable suggestion. We have removed this sentence.

Round 2

Reviewer 2 Report

Dear authors, 

the manuscript has improved greatly, you have answered every single question impeccably.

Kudos to you for your excellent work.